# An Integrated Time Series Prediction Model Based on Empirical Mode Decomposition and Two Attention Mechanisms

**Xianchang Wang [1], Siyu Dong [1] and Rui Zhang [2,*]**

[1] College of Computer Science and Technology, Jilin University, Changchun 130012, China; xcwang89@jlu.edu.cn (X.W.); dongsy21@jlu.edu.cn (S.D.)
[2] Key Laboratory of Symbol Computation and Knowledge Engineering of Ministry of Education, College of Computer Science and Technology, Jilin University, Changchun 130012, China
[*] Correspondence: rui@jlu.edu.cn

**Abstract:** In the prediction of time series, Empirical Mode Decomposition (EMD) generates subsequences and separates short-term tendencies from long-term ones. However, a single prediction model, including attention mechanism, has varying effects on each subsequence. To accurately capture the regularities of subsequences using an attention mechanism, we propose an integrated model for time series prediction based on signal decomposition and two attention mechanisms. This model combines the results of three networks—LSTM, LSTM-self-attention, and LSTM-temporal attention—all trained using subsequences obtained from EMD. Additionally, since previous research on EMD has been limited to single series analysis, this paper includes multiple series by employing two data pre-processing methods: 'overall normalization' and 'respective normalization'. Experimental results on various datasets demonstrate that compared to models without attention mechanisms, temporal attention improves the prediction accuracy of short- and medium-term decomposed series by 15~28% and 45~72%, respectively; furthermore, it reduces the overall prediction error by 10~17%. The integrated model with temporal attention achieves a reduction in error of approximately 0.3%, primarily when compared to models utilizing only general forms of attention mechanisms. Moreover, after normalizing multiple series separately, the predictive performance is equivalent to that achieved for individual series.

**Keywords:** time-series prediction; empirical mode decomposition; attention mechanism; data normalization

## 1. Introduction

Time series refers to a sequence of data points that vary over time, which widely exists in economic, industrial, and other sectors. Time-series prediction is the prediction of future trends at a specific time point or interval, derived from an established set of sequence observations, which serves as a guiding principle for making informed decisions about future actions. For example, investors frequently analyze recent stock price fluctuations to anticipate future movements. In recent years, there have been comprehensive studies on time-series prediction involving various machine-learning techniques. Among these, neural networks have been demonstrated to yield generally superior prediction outcomes compared to traditional machine-learning methods [1,2].

Because of the differences in time series themselves, they may exhibit various patterns of change, such as periodic, stability, and non-stability [3]. General time series is not only unstable in the trend but also has obvious fluctuation at the micro level, i.e., the change of data has strong uncertainty between adjacent sampling time points. Stock prices as well as sales, visitor flow, etc., belong to this type. Compared with others, these nonlinear and non-stationary sequences have more complex features, so that traditional machine-learning and deep-learning models have a larger prediction error [4]. Sequence stabilization, such as Fourier transform, wavelet transform, and Empirical Mode Decomposition (EMD), is a

common method used to deal with such sequences as those above. In recent years, EMD has been widely used in the prediction of this type of sequence. Through EMD, the short-term change and the long-term trend can be separated from the original sequences to improve the prediction effect.

However, considering the dataset selection, current studies of EMD have been conducted on a single sequence with many time nodes for each time. In fact, most datasets are composed of more than one sequence obtained from relatively independent entities of a single system, which are called "multiple sequences" in this paper. Although these sequences have the same attributes, there are many differences in value ranges and variation characteristics among them. As for the application of EMD to multiple sequences, there is still a lack of relevant research.

On the other hand, most studies use the same prediction model for the decomposed sequences. However, since the different subsequences have different characteristics and influence on the original sequence, the effect of a single model is different among each subsequence as well. Multiple models are sometimes considered in actual research. Consequently, we considered inputting each subsequence into these models, then selecting the model with the best prediction effect for each for integration, which is the idea of "decomposition–prediction–integration" [4].

The attention mechanism, initially developed for natural language processing, is a neural network model that can be applied to extract a set of feature vectors from any given problem. Consequently, the general attention model demonstrates its versatility across various domains, including time-series analysis [5]. It can be employed in conjunction with other neural networks, such as the Transformer [6], to predict time series.

Most studies regarding attention mechanisms with sequence decomposition are still limited to using the same model for each subsequence. However, among the subsequences, there is still the complexity of local features and the differences of global features, such as period, to be considered. Consequently, our focus is on selecting the most suitable integration model based on the prediction performance of each subsequence to enhance the overall prediction efficacy, in line with the "decomposition–prediction–integration" concept.

To summarize, this research presents an integrated time-series prediction model constructed upon EMD with two attention mechanisms, namely Self-Attention (SA, a.k.a. partial attention) and Temporal Attention (TA, a.k.a. global attention). The primary innovations of our model are as follows:

1. Employing the concept of "disintegration–prediction–integration", subsequences decomposed by CEEMDAN are separately input into three networks, namely LSTM, LSTM-Self-Attention (LSTM-SA), and LSTM-Temporal Attention (LSTM-TA), for training, followed by the selection of the optimal model for each subsequence to integrate.
2. Experiments were conducted on both single sequence and multiple sequence datasets. In addition, considering the characteristics of multiple sequences, two data preprocessing methods, "global normalization" and "separate normalization", are investigated and compared.

## 2. Literature Review

### 2.1. Empirical Mode Decomposition

The basic idea of Empirical Mode Decomposition (EMD) [7] is to decompose a nonlinear and non-stationary sequence into some subsequences with various period features. Since EMD is subject to mode aliasing, generation of false components, and terminal effects, Ensemble Empirical Mode Decomposition (EEMD) [8] has been proposed. This method involves the addition of a pair of positively and negatively correlated Gaussian white noise sequences to the pre-decomposed sequence. Compared to EEMD, Complete Ensemble Empirical Mode Decomposition with Adaptive Noise (CEEMDAN) [9] addresses the issue of noise transmission from high- to low-frequency.

EMD has been widely used in signal processing. Zheng et al. [10] improved the Uniform Phase Empirical Mode Decomposition (UPEMD) and applied it to rolling bearing

and rotor rubbing fault diagnosis. Adam et al. [11] applied EEMD to the exchange rate data in order to analyze similarities in the Southern African Development Community (SADC) exchange rate market's structure. Mousavi et al. [12] applied CEEMDAN to the structural damage detection of steel truss bridges, proving that the detection effect when using CEEMDAN is significantly better than that when using EMD or EEMD. EMD is also used in composite fault diagnosis of gearboxes [13], mechanical fault diagnosis [14], milling chatter detection [15], etc.

In time-series prediction, the prediction accuracy can be enhanced by introducing EMD into the learning model. Nguyen et al. [16] combined EEMD and Long Short-Term Memory (LSTM) to predict time-series signals in nuclear power plants. Peng et al. [17] combined CEEMDAN with the permutation entropy method and used the Convolution-based Gated Recurrent Neural Network (ConvGRU) to predict South Asian high intensity. Jin et al. [18] proposed a deep-learning model based on EMD with a back-propagation (BP) neural network and Particle Swarm Optimization (PSO). Guo et al. [19] proposed a model based on EMD, multi-view learning, and a winner-takes-all strategy. EMD is also used for Air Quality Index (AQI) forecasting with broad learning systems [20], tourism forecasting with the Recurrent Neural Network (RNN) [21], stock price prediction with the hybrid model of the Convolutional Neural Network (CNN), LSTM [22], etc.

Although EMD is widely used in many types of time series, in the existing researches, what is decomposed in each experiment is still a single original sequence. Sometimes the sequences to be processed may be multiple with the same properties but relative independence. Considering the differences of value ranges and variation characteristics, it is not necessarily appropriate to either decompose only one of the sequences or to simply concatenate each sequence before decomposition. Consequently, how to pre-process multiple sequences before decomposition is one focus of our research.

On the other hand, in each research regarding time-series prediction with EMD mentioned above, the same learning model is applied in all decomposed sequences. The comparison of multiple models is still limited to the overall effect. Due to the difference of periodicity, it can be observed that, although one model demonstrates optimal prediction performance on some subsequences, other distinct models work best on other subsequences. Consequently, when evaluating multiple learning models in a model based on EMD, it is necessary to consider the performance diversity of these models within the subsequences, in addition to comparing the overall effect of them.

*2.2. Attention Mechanism*

The earliest work on attention mechanisms is Bahdanau et al.'s machine translation model [23], which addressed certain issues with RNN's structure. As attention mechanisms have been used in multiple fields related to text and image processing, they have gradually become popular in deep learning [5]. Moreover, the introduction of Transformer [6] further proves the effectiveness of attention. Attention is widely applied in many types of subsequences, such as text, audio, video, and of course, time series.

This section focuses on attention's application of time series. Li et al. [24] combined LSTM with attention to predict water conservancy data. Hu J. et al. [25] designed a multistage attention network with influential attention and temporal attention to study the influence of different non-predictive time series on target series in different time stages in historical data. He et al. [26] proposed an encoder–decoder network based on dual attention enhancement and LSTM for typhoon track prediction. Hu Y.-T. et al. [27] proposed a RNN based on network self-attention to study similarity scores in time series. Lai et al. [1] proposed a Long- and Short-term Time-series network (LSTNet), which used temporal attention mechanisms to learn from the sequences with uncertain periods. Based on LSTNet, Wang et al. [28] introduced spatial and temporal self-attention to discover the dependences between variables and the relationships among historical observations.

Transformer [6] is a sequence-to-sequence model containing multiple tensors of multihead self-attention mechanisms. Since the $O(L^2)$ ($L$ is the length of the input window

sequence) complexity influences Transformer's performance of learning long sequences, many improvements of it have been made, such as LogTrans [29], Performer [30], Reformer [31], Informer [32], Autoformer [33], Fedformer [34], Scaleformer [35], etc., most of which have $O(L \log L)$ complexity.

In terms of a combination of attention with EMD, Chen et al. [36] proposed a hybrid model consisting of EMD and the attention-based LSTM. Hu Z.-D. [37] proposed a prediction model based on CEEMDAN and LSTM with an attention mechanism and constructed a news sentiment index based on news texts to predict crude oil prices. Neeraj et al. [38] used EMD to compose electric load data and proposed an encoder–decoder model with LSTM and attention for prediction. In Huang et al.'s short-term metro passenger flow prediction [39], the series after CEEMDAN are reconstructed and trained in the attention-based Seq2Seq model. Yu et al. [40] combined and improved EEMD with Variational Mode Decomposition (VMD) to decompose photovoltaic power series and predict the subsequences by bidirectional LSTM with the whale optimization algorithm and attention mechanism. However, most of the existing studies using attention in the model based on EMD are still limited in applying attention to all subsequences (including reconstructed subsequences) simultaneously, which is the same as what has been stated in Section 2.1.

Liu et al. [41] proposed a hybrid model containing EEMD, entropy-based denoising, GRU, and history attention to predict stock prices. After denoising, the subsequences are reconstructed as high-, medium-, and low-frequency. GRU with History Attention (GRU-HA) is used to train reconstructed high- and medium-frequency subsequences, while BP is used to train reconstructed low-frequency subsequences. However, due to the lack of sufficient benchmarks, the paper only shows, from the experiment result, that BP achieved better prediction effects than GRU-HA on reconstructed low-frequency subsequence. On low-frequency subsequence, to further prove that the attention mechanism could not obtain better prediction effects, effect comparison between GRUs with and without attention is necessary. Meanwhile, there is a lack of clear selection basis of prediction models for each subsequence.

To sum up, when the attention mechanism is applied in the time series after decomposition, most papers still apply the whole prediction model to all subsequences. Even if there is individual research that applies different prediction models to the subsequences, it considers only the comparison among the whole model without the comparison on each subsequence among multiple prediction models.

## 3. Preliminaries

### 3.1. Empirical Mode Decomposition

EMD is a method for processing nonlinear and non-stationary sequences, which can effectively separate different period features of the sequence. The basic idea of EMD is to decompose an original sequence into a series of Intrinsic Mode Function (IMF) subsequences and a rest sequence. The procedure for EMD is as follows:

1. For the original sequence $x(t)$, set $r(t) = x(t)$ and $k = 1$.
2. Set $m(t) = r(t)$.
3. Cubic spline interpolation was used to fit all local minima (respective maxima), ending up with an envelope, $e_{\min}(t)$ (respective $e_{\max}(t)$).
4. Calculate $m(t)$ and $c(t)$:

$$m(t) = \frac{1}{2}(e_{\min}(t) + e_{\max}(t)) \tag{1}$$

$$c(t) = r(t) - m(t) \tag{2}$$

5. If $c(t)$ meets the conditions of IMF as in the following, set $\text{IMF}_k(t) = c(t)$ as an IMF subsequence of $x(t)$ and

$$r(t) = x(t) - c(t): \tag{3}$$

a.  For the entire data set, the number of extrema and the number of zero crossings must either be equal or differ, at most by one;

b.  At any point, the mean value of the envelopes defined by the local maxima and the local minima must be zero.

Otherwise, let $m(t) = c(t)$ and repeat Steps 3~5.

6.  IF $r(t)$ is a constant sequence or it includes at most one minimum point and one maximum point each (in current studies, the termination condition of the EMD algorithm is usually expressed as $r(t)$ and is a constant or monotone sequence. However, the series with at most one minimum point and one maximum point is still unable to be decomposed), which means it is undecomposable, this is the end of the algorithm. In this case, set $\text{RES}(t) = r(t)$ as the remaining sequence; we then obtain $k$ IMF subsequences $\text{IMF}_i(t)$ $(i = 1, 2, \ldots, k)$ and a remaining sequence, $\text{RES}(t)$, i.e.,

$$x(t) = \sum_{i=1}^{k} \text{IMF}_i(t) + \text{RES}(t) \tag{4}$$

Otherwise, let $k = k + 1$ and repeat Steps 2~6.

EEMD and CEEMDAN are EMD's two most common variants. We used CEEMDAN to decompose the sequences in this paper. The procedure for CEEMDAN is as follows:

1.  For the original sequence $x(t)$, set $r(t) = x(t)$ and $k = 1$.
2.  For $N$ times, add Gaussian white noise sequences to $r(t)$, obtaining the following $N$ sequences:

$$x_{k,j}(t) = \begin{cases} r(t) + n_{k,1}(t), & j = 1 \\ x_{k,j-1}(t) + n_{k,j}(t), & j = 2, 3, \ldots, N \end{cases} \tag{5}$$

where $n_{k,j}(t)(j = 1, 2, \ldots, N)$ is the Gaussian white noise sequences in the $j$th trial and $x_{k,j}(t)$ is the time series with the additional noise.

3.  EMD is used to decompose $x_{k,j}(t)$. Extract the first IMF subsequence of each sequences' decomposition results, and the average of them becomes the first subsequence of the final decomposition as:

$$\text{IMF}_k(t) = \frac{1}{N} \sum_{j=1}^{N} \text{imf}_1\left(x_{k,j}(t)\right) \tag{6}$$

where $\text{imf}_1(\cdot)$ is a function that obtains the first IMF subsequence of a sequence through EMD.

4.  The remaining sequence is:

$$r(t) = x(t) - \text{IMF}_k(t) \tag{7}$$

5.  IF $r(t)$ is a constant sequence or it includes at most one minimum point and one maximum point both, which means it is undecomposable, this is the end of the algorithm. In this case, set $\text{RES}(t) = r(t)$ as the remaining sequence, then we obtain $k$ MF subsequences $\text{IMF}_i(t)$ $(i = 1, 2, \ldots, k)$ and a remaining sequence, $\text{RES}(t)$, i.e.,

$$x(t) = \sum_{i=1}^{k} \text{IMF}_i(t) + \text{RES}(t) \tag{8}$$

Otherwise, let $k = k + 1$ and repeat Steps 2~5.

EMD, EEMD, and CEEMDAN can be performed by calling the EMD-signal/PyEMD ("EMD-signal" is for install and "PyEMD" is for import in code. The library is not pre-installed in Python, Anaconda, or Pytorch, and should be installed by users) library in Python.

Research has shown that there is little diversity of prediction effect between using EMD and CEEMDAN in time-series prediction [6]. Since solving EMD's disadvantages of mode aliasing, generation of false components, and terminal effects, the subsequences from CEEMDAN can express the regularity of the original sequence in different periods more accurately. However, in the usual case, the time complexity of CEEMDAN is $O(Nn\log n)$, while that of EMD is $O(n\log n)$, where $n$ is the length of the sequence and $N$ is the times of adding noise sequences. In the EMD-signal/PyEMD library of Python, the default value of $N$ is 100, which makes the time cost of CEEMDAN significantly higher than that of EMD.

*3.2. LSTM*

The Recurrent Neural Network (RNN) is a type of neural network model used exclusively for the analysis and prediction of time series. Since gradient disappearance and explosion make it difficult for traditional RNNs to learn long-distance dependencies, gates are added to the RNN to form LSTM [42] and GRU (Gated Recurrent Unit) [43]. The two networks selectively remember and forget information through the gates to learn the long-term correlation features of the sequence, which effectively solves the long-term dependence problem. All three of the above networks are widely adopted time-series prediction models.

We selected LSTM for our work. The LSTM unit is composed of three gate layers, as shown in Figure 1, forget gate, input gate, and output gate, through which the selective memory and long-term dependence on input information are realized.

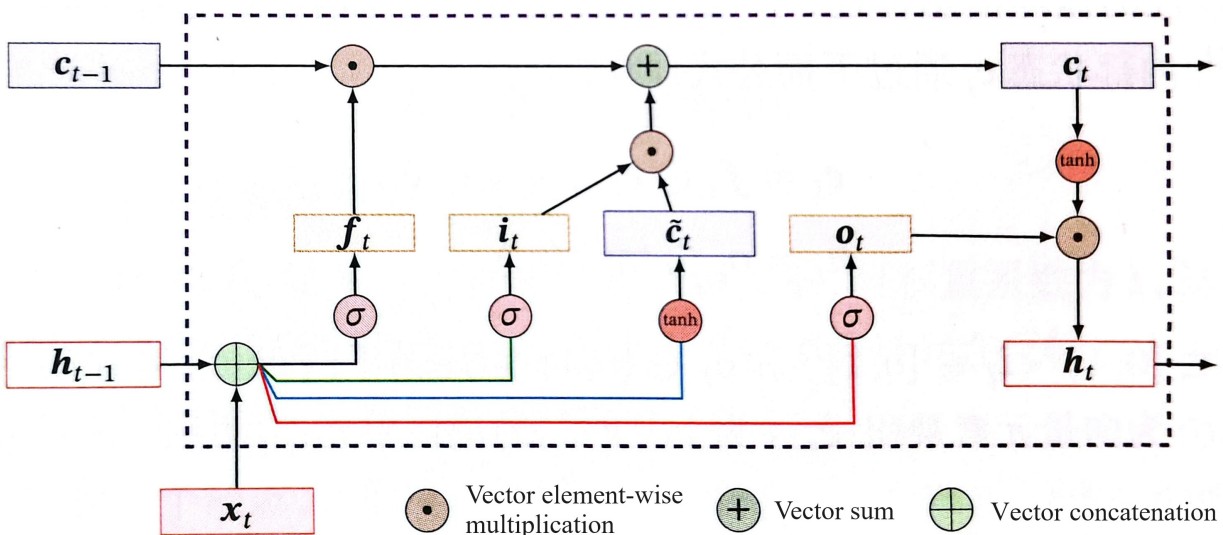

**Figure 1.** Basic structure of LSTM.

The algorithm of LSTM can be expressed by the following [44]:

$$\begin{bmatrix} \tilde{c}_t \\ o_t \\ i_t \\ f_t \end{bmatrix} = \begin{bmatrix} \tanh \\ \sigma \\ \sigma \\ \sigma \end{bmatrix} \left( W \begin{bmatrix} x_t \\ h_{t-1} \end{bmatrix} + b \right) \tag{9}$$

$$c_t = f_t \odot c_{t-1} + i_t \odot \tilde{c}_t \tag{10}$$

$$h_t = o_t \odot \tanh(c_t) \tag{11}$$

where $x_t$ is the input for the current moment, $W$ and $b$ are parameters to be learnt, $\sigma$ is the sigmoid function, and $\odot$ is the product of the corresponding elements to the tensor.

*3.3. Attention Mechanism*

The attention mechanism is a method that imitates the human visual nerve and uses limited computing resources to process more important information. According to the usage, attention can be divided into temporal attention, spatial (feature) attention, etc. It is often used in conjunction with encoder–decoder frameworks such as Transformer [6].

The general attention mechanism can be represented as a key-value pair, as shown in Figure 2. Firstly, compute the attention distribution $A = \{\alpha_{ij}\}$ between tensors $K$ and $Q$:

$$\alpha_{ij} = \text{softmax}(s(k_j, q_i)) = \frac{\exp(s(k_j, q_i))}{\sum_{m=1}^{N} \exp(s(k_m, q_i))} \quad (12)$$

where $N$ is the length of the input sequence, $i$, $j \in \{1, 2, \ldots, N\}$ are, respectively, the positions of the output and the input sequences, $k_j$, $q_i$ are, respectively, the $j$th row vector of $K$ and the $i$th row vector of $Q$, and $s(\cdot)$ is the Attention Scoring Function (ASF), including scaled dot product, cosine similarity, etc. The scaled dot product is used in this paper.

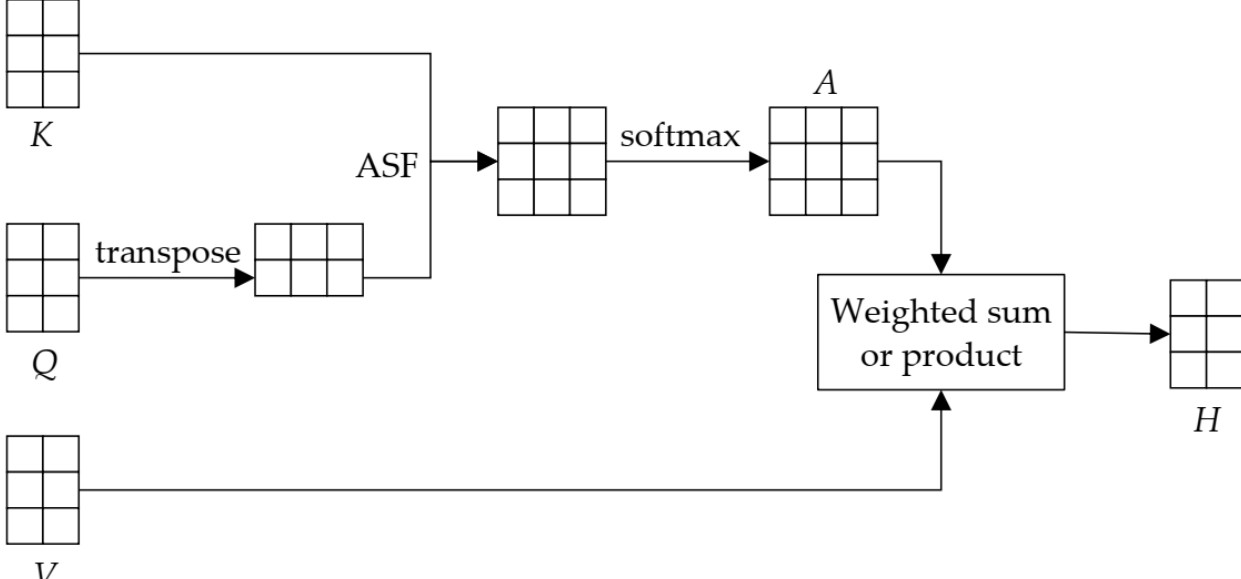

**Figure 2.** Basic structure of attention mechanism.

There are two ways to output $H$. One way is by calculating the weighted sum of tensor $V$ based on tensor $A$ as:

$$h_i = \sum_{j=1}^{N} \alpha_{ij} v_j \quad (13)$$

where $h_i$ is the $i$th row vector of $H$ and $v_j$ is the $j$th row vector of $V$.

The other way, which is mainly used in self-attention, is by multiplying tensor $V$ with tensor $A$ as:

$$H = A \times V \quad (14)$$

## 4. Methodology

*4.1. General Framework*

For the proposed integrated model of time-series prediction based on CEEMDAN and two types of LSTM-attention, the framework is shown in Figure 3. The model adopts the idea of "decomposing–prediction–integration". The original sequence is normalized and decomposed into several subsequences, including a set of IMF sequences, $\text{IMF}_1$, $\text{IMF}_2$, . . ., $\text{IMF}_n$, and a remaining sequence, RES. Each subsequence is input to LSTM, LSTM-Self-Attention (LSTM-SA), or LSTM-Temporal Attention (LSTM-TA) for training. The better

model output is selected as the output of each subsequence, and the integration of these output sequences is output as the final prediction result.

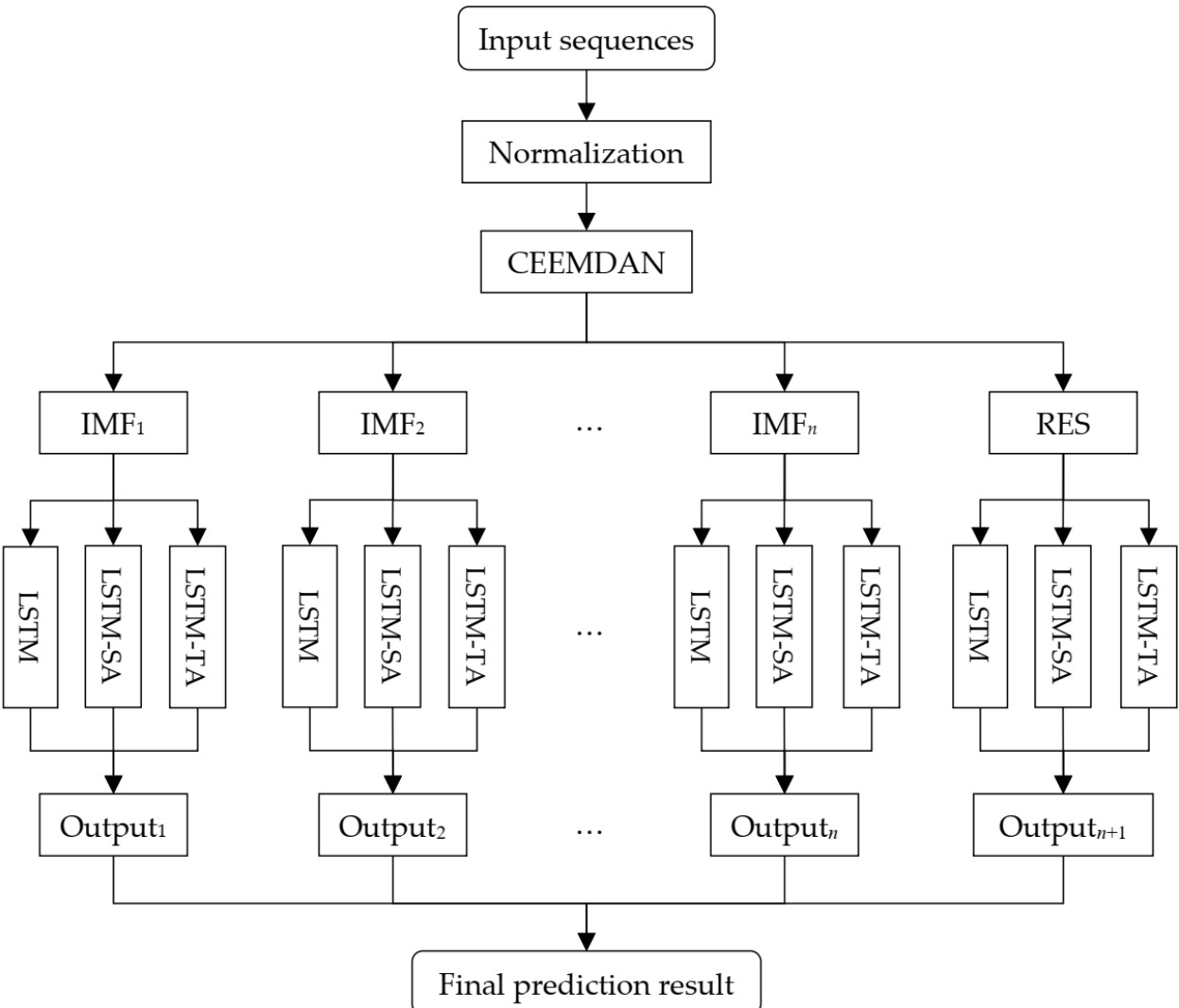

**Figure 3.** Framework of our model.

### 4.2. Single Sequence and Multiple Sequence

For the convenience of description, we provide the following definition of "single sequence" and "multiple sequence".

Time series does not exist in isolation. It shows the change of one or several attributes of one or a class of subjects in time. In a variety of time series, some reflect only one subject, which are named "single sequence". In contrast, the other datasets reflect two or more subjects that are relatively independent from one other, i.e., changes in the data of one subject do not make any change in the other. We refer to such time series containing multiple subjects as "multiple sequence" or "multi-sequence".

It should be noted that "multiple sequence" is different from "multivariate sequence" and "multi-dimensional sequence". "Multiple sequence" is a sequence that has the same attributes for multiple subjects, as opposed to only one subject, while the other two, a.k.a. "tensor sequence" and "tensor time series" [45], are sequences that represent multiple attributes, as opposed to only one attribute, for each subject.

*4.3. Data Pre-Processing: Global and Separate Normalization*

Normalization aims to scale the data by a certain rule to narrow the gap of the data magnitude. The common two normalization methods are the min–max scaler and Z-normalization (i.e., the normalization according to the mean and standard deviation). Given that the min–max scaler is used in most of the existing research, we use it in the paper.

For single sequences, the whole dataset can be directly normalized. For multiple sequences, there are two ways of normalization shown below.

- Global normalization

Each sequence segment is concatenated along the time dimension and normalized as a whole. For example, there are two sequences:

$$\{a_n\} = \{0, 1, 2, 3, 4\}$$

$$\{b_n\} = \{6, 7, 8, 9, 10\}$$

After the global min–max scaler, the two sequences become:

$$\left\{a_n'\right\} = \{0, 0.1, 0.2, 0.3, 0.4\}$$

$$\left\{b_n'\right\} = \{0.6, 0.7, 0.8, 0.9, 1\}$$

However, in multiple sequences, there may possibly be large diversity in the range of values covered by the individual sequence segments (e.g., the price of one stock is tens to hundreds of dollars, while the price of another is only a few dollars). If they are normalized globally, this diversity will not be eliminated, which will still increase the difficulty and reduce the efficiency of neural network training, thus affecting the accuracy of prediction. To eliminate this diversity, there is the following method, named separate normalization.

- Separate normalization

Each sequence segment is normalized separately and then concatenated along the time dimension for further research. For the sequences $\{a_n\}$ and $\{b_n\}$ above, after the separate min–max scaler, they will be as follows:

$$\left\{a_n'\right\} = \{0, 0.25, 0.5, 0.75, 1\}$$

$$\left\{b_n'\right\} = \{0, 0.25, 0.5, 0.75, 1\}$$

Obviously, this method eliminates the diversity of the range among the sequence segments. However, in order to denormalize accurately before outputting, it is necessary to record the length of each segment in the order of sequence segment splicing as well as determine the maximum and minimum value of each segment, respectively, for subsequent denormalization.

*4.4. Processing of Decomposed Subsequences*

In the model based on EMDs, even if the original data have been normalized, the value range of decomposed subsequences obtained will change. For instance, Figure 4 shows the curves of the NYtem dataset (see Section 5.1) and its two subsequences after CEEMDAN. Though the input sequence has been normalized, the ranges of decomposed sequences are different from the sequence before decomposition.

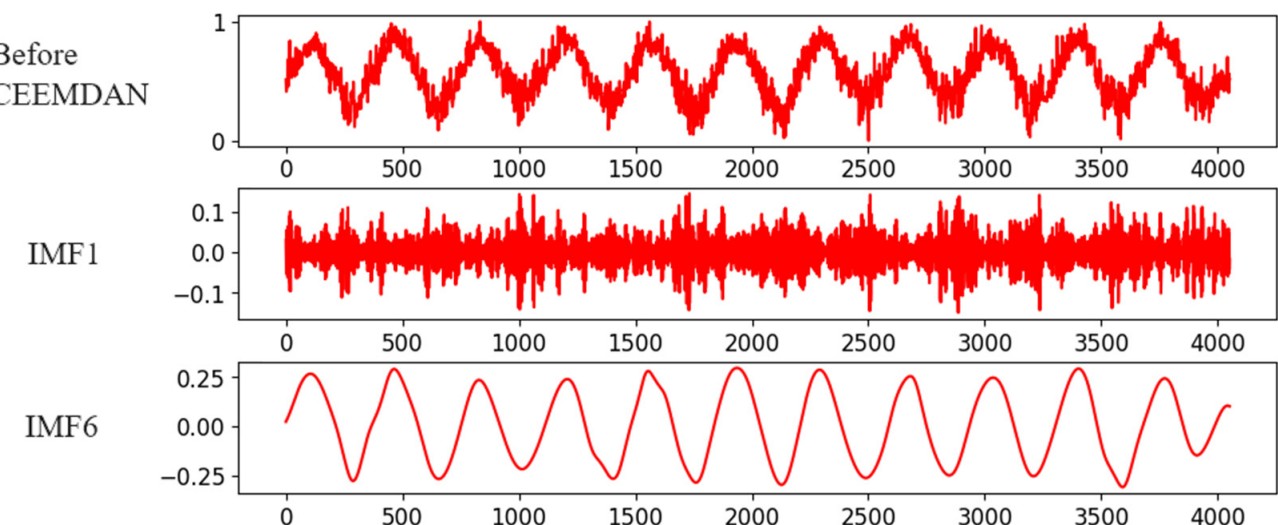

**Figure 4.** The images of the normalized NYtem dataset and its two subsequences after CEEMDAN.

Neural networks are sensitive to the value of the input data, i.e., either a too-large or a too-small value will lead to a small gradient, which makes training difficult. If the range of the subsequence is very small, the convergence of the neural network's gradient will be affected. Consequently, it is necessary to normalize each decomposed subsequence again. In this case, normalization is unnecessary for single sequences before decomposition. For multiple sequences, we need not scale to the interval [0, 1] during the separate normalization before decomposition, but the interval should not be too large or too small.

### 4.5. Two Attention Mechanisms

The neural network model based on EMD involves decomposing each subsequence into neural network models for prediction. The prediction results of each subsequence are then aggregated as the output. During training, each subsequence functions as a relatively independent entity, allowing for the use of distinct neural network models. To achieve optimal overall performance, we can select the most appropriate model for each subsequence based on the training effectiveness of each model. This concept is referred to as the "decomposition–prediction–integration" strategy [4].

In this study, we have employed three models: LSTM, LSTM-Self-Attention (LSTM-SA), and LSTM-Temporal Attention (LSTM-TA). LSTM-SA and LSTM-TA are both encoder–decoder models, with LSTM serving as the encoder and the attention mechanism serving as the decoder. The output of LSTM is utilized as the input for the attention mechanism. By implementing CEEMDAN on each subsequence and inputting the results into the three models, we can compare their respective effects and choose the model with the better performance for integration. LSTM has been described in Section 3.2; thus, we will only discuss the two attention mechanisms in this section.

#### 4.5.1. Self-Attention (Partial Attention) Mechanism

The Self-Attention (SA) mechanism is a model designed to capture the internal interdependency of sequences, commonly employed in text analysis and time-series prediction. Since SA operates only on the input window sequence of LSTM (i.e., a small sequence used for prediction), it can also be named "partial attention" in time-series prediction.

After sequence decomposition, the short-term regular subsequence reflects the short-term oscillation characteristics of the original sequences. Besides the stationary nature of the subsequence itself, its period is shorter than the length of the LSTM input window. Consequently, compared to the original sequence with more complex features and the medium- and long-term regular subsequence, whose period is longer than the

length of the input window, SA is more advantageous for extracting internal features in short-period subsequences.

In contrast to the Transformer, which also relies on self-attention, the self-attention employed in this study consists of only a single tensor of single-head attention. Specifically, in the Transformer encoder, set both the number of heads and the number of layers to 1 and eliminate the feed-forward layer to obtain the self-attention utilized in this work.

The process of SA is shown in Figure 5. For the given tensor $X \in R^{T \times D}$, using the scaled dot product as ASF, the calculation procedure is as follows:

$$Q = XW_q \in R^{T \times D} \tag{15}$$

$$K = XW_k \in R^{T \times D} \tag{16}$$

$$V = XW_v \in R^{T \times D} \tag{17}$$

$$Atten = \text{softmax}\left(\frac{KQ^{\mathrm{T}}}{\sqrt{T}}\right)V \tag{18}$$

where $T$ is the inputted time step length; $D$ is the output dimension of LSTM; and $W_q \in R^{D \times D}$, $W_k \in R^{D \times D}$, and $W_v \in R^{D \times D}$ are the parameter matrixes of linear mapping.

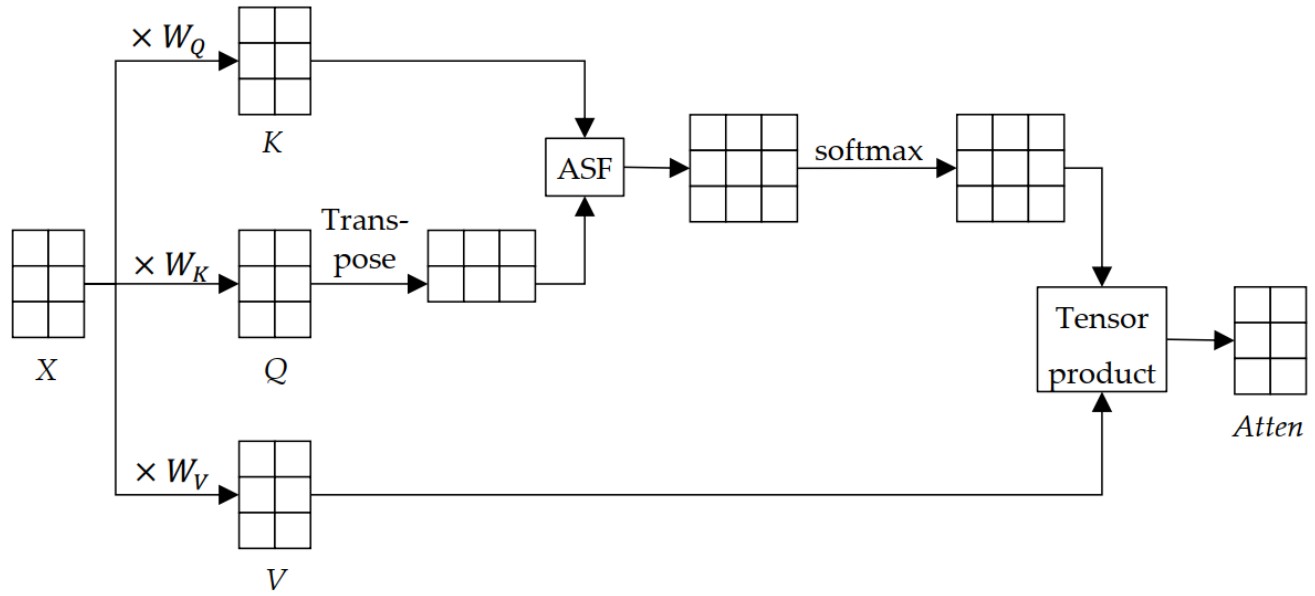

**Figure 5.** Procedure of self-attention ($T = 3$, $D = 2$).

### 4.5.2. Temporal (Global) Attention Mechanism

LSTM has solved the long-term dependence problem of RNN; however, "long short-term memory" is different from "long-term memory". When the whole sequence is relatively long or the length of the input sequence is uncertain, the information storing ability of the hidden and the cell state output by LSTM is still limited.

The Temporal Attention (TA) mechanism, a.k.a. global attention mechanism, is one of the methods used to better capture the rules of sequences in a long time period. Different from SA, which only focuses on a small sequence, it is a neural network model able to capture long-term patterns in the whole sequence. Through EMDs, the original sequence produces a set of subsequences with different periods. The purpose of TA is to learn the periodicity of the whole sequence over a long period of time. It is more beneficial to capture the periodic regularity of subsequences whose period is significantly shorter than the whole

sequence length (the specific degree varies depending on the specific dataset, which is generally shorter than 1/10 of the whole length). What is more, since each subsequence is a stable sequence with periodic oscillations, the periodic rule is more obvious than the non-stationary original sequence with complex changing trend, which is more beneficial to learning the periodic character through TA.

The main process of TA is shown in Figure 6 and described as follows:

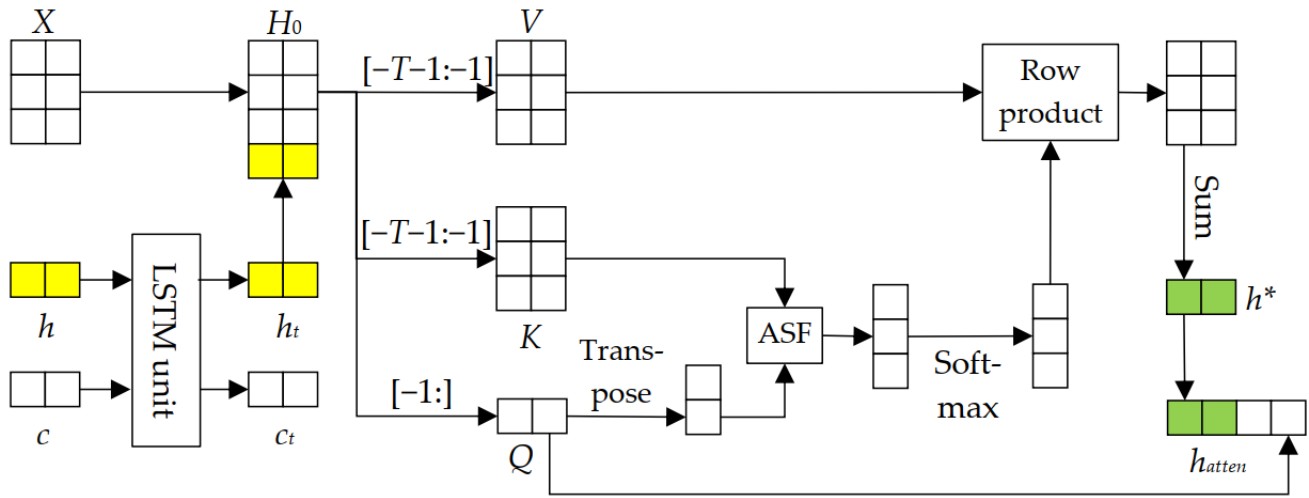

**Figure 6.** Procedure of temporal attention ($T = 3$, $D = 2$).

1. The hidden state, $h$, and the cell state, $c$, output by the LSTM are input to the LSTM unit for one-step prediction, obtaining the updated hidden state $h_t$ and cell state $c_t$.
2. Concatenate $h_t$ to the tail of $X$, the output tensor of the LSTM encoder, along the time dimension as the prediction sequence, $H_0$.
3. Let $Q$ be the last time step of $H_0$ and $K$ and $V$ be the $(T + 1)$th to the 2nd time step from the bottom of $H_0$, where $T$ is the time dimension of the input to the LSTM encoder. Calculate the attention value according to the equation below:

$$Atten = \text{diag}\left( \text{softmax}\left( \frac{KQ^{\text{T}}}{\sqrt{T}} \right)^{\text{T}} \right) V \tag{19}$$

where "diag(·)" is a function to convert the column vector to a diagonal matrix. In this case, Equation (19) is equivalent to multiplying each dimension of the column vector output by the softmax function with the corresponding row vector in $V$, which is the "row product" described in Figure 6.

4. Add the attention values for each time dimension in *Atten*, i.e.,

$$h^* = \sum_{i=1}^{T} atten_i \tag{20}$$

where $atten_i$ is a vector in the $i$th time dimension of *Atten*.

5. Concatenate $Q$ to the tail of $h^*$, which becomes the prediction result $h_{atten}$ to output.

TA can be used for multi-step prediction by looping the above procedure in the decoder, setting $h = h_t$ and $c = c_t$ after each step, and concatenating the prediction results of each step along the time dimension in turn. Considering the experimental requirements, multi-step prediction is not discussed in this paper.

### 4.5.3. Comparison and Limitation of the Two Attention Mechanisms

As mentioned above, SA is suitable for predicting short-term regularity, while TA has the advantage of predicting short- and medium-period regularities. Besides that, the effect diversity between the two is reflected in the structure.

SA only accepts the output tensor of LSTM as its input without using the historical information in the hidden state, $h$, or the cell state, $c$, so it is limited to the extraction of regularity inside the input window. TA processes the hidden state, $h$, of the LSTM output and lets it become the query of attention mechanism, fully utilizing the historical information stored in $h$ during the process of extracting features. Consequently, TA can not only extract periodic regularities longer than the input window, but also extract short-period regularities more accurately than SA.

However, not all subsequences are applicable to attention mechanisms. TA's ability of promoting LSTM's memory is still limited. The last few subsequences generated through EMD have a long period, which is usually hundreds of times that of the input window, which make it difficult for TA to memorize this long-period regularity. To achieve better overall results, we considered using multiple network models to train and select the appropriate one for each subsequence according to the effect, as in the whole model we designed in Section 4.1.

## 5. Experiments

### 5.1. Datasets and Experiment Environment

The following three datasets from Kaggle will be used for the experiment.

- New York Daily Average Temperature (NYtem): From a large database containing the daily average temperature data of 321 cities in 125 countries since 1995, the data of New York City from 10 April 2009 to 13 May 2020 are selected under the preconditions, including avoiding missing values, with a total of 4052 entries. The prediction rule is to use 10 consecutive data items to predict the next data.
- Monthly ReTail Sales of the USA (MRTS): It contains the data of monthly sales in various fields of the US retail industry from January 1992 to May 2020. We selected the original statistical data stored in ".xls" format. Eliminating some total items and combining with some situations in the experiment, 8882 data items in 28 fields were selected. Six consecutive items were used to predict the next.
- SocioEconomic Status Score (SES): It contains 2086 socioeconomic status percentage scores for 149 countries every 10 years between 1880 and 2010. We used six consecutive entries to predict the next one.

The three datasets above are time series with microscopic oscillation characteristics. To be specific, NYtem is a single sequence, while MRTS and SES are multiple sequences. Meanwhile, the length of each sequence part in MRTS is significantly larger than that in SES.

Experiment environment: Windows 10, Python 3.8, Pytorch 1.11 on CPU, installing EMD-signal 1.4 and other necessary third-party libraries. VScode and Jupyter Notebook are the development environment.

### 5.2. Methods for Comparison and Evaluation Criteria

We use the following methods in our comparative evaluation:

- For prediction tasks without sequence decomposition, besides LSTM, LSTM-SA, and LSTM-TA, we also used Support Vector Regression (SVR) with linear kernel function (SVR-linear), SVR with Radial Basis Function (a.k.a. Gaussian kernel function) (SVR-RBF), eXtreme Gradient Boosting (XGBoost), Light Gradient Boosting Machine (LightGBM), BP, CNN, RNN, and GRU.
- For prediction tasks with sequence decomposition and a single prediction model, besides LSTM, LSTM-SA, and LSTM-TA, we also used RNN and GRU.

- For prediction tasks with sequence decomposition and multiple prediction models, we considered three ways of model integration: (1) SA integration, our proposed integrated model with the selection between LSTM and LSTM-SA for each subsequence; (2) TA integration, our proposed integrated model with the selection among LSTM, LSTM-SA, and LSTM-TA for each subsequence; (3) RLG integration, an integrated model with the selection among RNN, LSTM, and GRU [4].

Four criteria were adopted in this study, including Mean Absolute Error (MAE), Root Mean Squared Error (RMSE), Mean Absolute Percentage Error (MAPE), and R-square ($R^2$).

MAE and RMSE both measure the absolute error between the prediction results and the real data, where RMSE is sensitive to the results with large errors.

$$\text{MAE} = \frac{1}{N} \sum_{i=1}^{N} |y_i - \hat{y}_i| \tag{21}$$

$$\text{RMSE} = \sqrt{\frac{1}{N} \sum_{i=1}^{N} (y_i - \hat{y}_i)^2} \tag{22}$$

MAPE measures the relative error between the predicted and the real data.

$$\text{MAPE} = \frac{1}{N} \sum_{i=1}^{N} \left| \frac{y_i - \hat{y}_i}{y_i} \right| \tag{23}$$

To be consistent with the representation of the experimental results, "$\times 100\%$" in the formula described in most papers is deleted. This is because the output of the "mean_absolute_percentage_error" function in the "sklearn.metrics" library of Python is expressed as a decimal rather than a percentage (e.g., 90% is expressed as 0.9).

$R^2$ measures how well the model fits on the whole dataset. When its value is closer to 1, the model fits better.

$$R^2 = 1 - \frac{\sum_{i=1}^{N} (y_i - \hat{y}_i)^2}{\sum_{i=1}^{N} (y_i - \overline{y})^2} \tag{24}$$

In Equations (21)~(24), $y_i$ is true value in the original sequence, $\hat{y}_i$ is the predicted value, and $\overline{y} = \sum_{i=1}^{N} y_i$.

### 5.3. Experiments on Dataset NYtem

The normalized NYtem datasets is input into CEEMDAN. Each decomposed subsequence is input to LSTM without and with secondary normalization, respectively, for comparison. Determining whether secondary normalization is needed, each subsequence is input into LSTM-self-attention and LSTM-temporal attention. Comparing predicting effects with LSTM, the optimal integrated model is selected to compare with the undecomposed model and the single model with decomposition.

After sequence decomposition, eight IMF subsequences and a residual-term RES are obtained. These sequences are secondarily normalized and input into LSTM with the unnormalized sequences, respectively, with the comparison of $R^2$ indexes. The results are shown in Table 1.

**Table 1.** Predicting effect comparison of LSTM on NYtem between the subsequences without and with secondary normalization.

| Subsequence | $R^2$ without Secondary Normalization | $R^2$ with Secondary Normalization |
|---|---|---|
| $IMF_1$ | 0.209830 | 0.226047 |
| $IMF_2$ | 0.915300 | 0.919399 |
| $IMF_3$ | 0.997985 | 0.997993 |
| $IMF_4$ | 0.999910 | 0.999810 |
| $IMF_5$ | 0.999945 | 0.999879 |
| $IMF_6$ | 0.999975 | 0.999956 |
| $IMF_7$ | 0.999161 | 0.999540 |
| $IMF_8$ | 0.998578 | 0.999868 |
| RES | $-4.207980$ | 0.999857 |

In Table 1, there is little diversity in the prediction effect between IMFs before and after secondary normalization. However, it shows serious overfitting on RES without the normalization. Accordingly, we believe that a secondary normalization is essential before inputting the subsequences into the neural network. Subsequences are all secondarily normalized in the follow-up experiments.

Table 2 shows the prediction effects of each decomposition subsequence on LSTM and two LSTMs with attention. Considering that the value of these subsequences may be close to or even equal to zero so that it is not suitable to use MAPE, only MAE, RMSE and $R^2$ criteria are listed in the table. In all the tables below, for each subsequence and evaluation index, the best results are marked in bold and the second best are underlined.

**Table 2.** Prediction results of LSTM, LSTM-SA, and LSTM-TA for each decomposed subsequence of NYtem.

| Subsequences | LSTM | | | LSTM-SA | | | LSTM-TA | | |
|---|---|---|---|---|---|---|---|---|---|
| | **MAE** | **RMSE** | $R^2$ | **MAE** | **RMSE** | $R^2$ | **MAE** | **RMSE** | $R^2$ |
| $IMF_1$ | 0.027933 | 0.035760 | 0.221853 | <u>0.027081</u> | <u>0.034681</u> | <u>0.268124</u> | **0.023834** | **0.032200** | **0.369078** |
| $IMF_2$ | 0.007363 | 0.010486 | 0.918929 | <u>0.007320</u> | <u>0.010011</u> | <u>0.926119</u> | **0.006302** | **0.008748** | **0.943575** |
| $IMF_3$ | <u>0.000999</u> | <u>0.001522</u> | <u>0.997991</u> | 0.003152 | 0.005126 | 0.977205 | **0.000953** | **0.001505** | **0.998036** |
| $IMF_4$ | <u>0.000293</u> | <u>0.000369</u> | <u>0.999808</u> | 0.001652 | 0.002133 | 0.993582 | **0.000132** | **0.000175** | **0.999957** |
| $IMF_5$ | **0.000202** | **0.000259** | **0.999884** | 0.001201 | 0.001677 | 0.995127 | <u>0.000507</u> | <u>0.000540</u> | <u>0.999495</u> |
| $IMF_6$ | <u>0.000895</u> | <u>0.001158</u> | <u>0.999955</u> | 0.005709 | 0.007149 | 0.998275 | **0.000443** | **0.000566** | **0.999989** |
| $IMF_7$ | **0.000892** | **0.001029** | **0.999538** | 0.002317 | 0.003177 | 0.995594 | <u>0.002020</u> | <u>0.002384</u> | <u>0.997518</u> |
| $IMF_8$ | **0.000213** | **0.000344** | **0.999875** | 0.000706 | 0.000937 | 0.999072 | <u>0.001254</u> | <u>0.001464</u> | <u>0.997735</u> |
| RES | **0.000035** | **0.000057** | **0.999830** | 0.000127 | 0.000163 | 0.998614 | <u>0.000057</u> | <u>0.000063</u> | <u>0.999793</u> |

In Table 2, compared with the model without attention, the prediction effect of SA on short-term regular subsequences, $IMF_1$ and $IMF_2$, is slightly improved, while TA achieves a much better effect on $IMF_1 \sim IMF_4$ and $IMF_6$ than the two model. According to Table 2, the following two integration models are selected:

- SA integration: LSTM-SA is used for $IMF_1$ and $IMF_2$, while LSTM is used for the others.
- TA integration: LSTM-TA is used for $IMF_1 \sim IMF_4$ and $IMF_6$, while LSTM is used for $IMF_5$, $IMF_7$, $IMF_8$, and RES.

All models mentioned in Section 5.2 are applied in this and the following experiments. Table 3 shows the results of each experiment.

**Table 3.** Results of the experiment on NYtem dataset.

| Type | Network Model | MAE | RMSE | MAPE | R$^2$ |
|------|--------------|-----|------|------|-------|
| Without CEEMDAN | SVR-linear | 3.913410 | 5.143536 | 0.082907 | 0.905441 |
| | SVR-RBF | 3.934860 | 5.164794 | 0.083349 | 0.904657 |
| | XGBoost | 4.073632 | 5.336617 | 0.087877 | 0.898208 |
| | LightGBM | 4.048071 | 5.328266 | 0.087172 | 0.868526 |
| | BP | 3.867268 | 5.132837 | 0.083532 | 0.905834 |
| | CNN | 3.862023 | 5.108569 | 0.082561 | 0.906722 |
| | RNN | 3.846566 | 5.100761 | 0.082990 | 0.907007 |
| | LSTM | 3.850042 | 5.093421 | 0.082190 | 0.907274 |
| | GRU | 3.848090 | 5.091207 | 0.082539 | 0.907354 |
| | LSTM-SA | 3.954155 | 5.211856 | 0.085696 | 0.902911 |
| | LSTM-TA | 3.833786 | 5.067739 | 0.081731 | 0.908325 |
| CEEMDAN + single network | RNN | 2.422320 | 3.136646 | 0.051033 | 0.964835 |
| | LSTM | 2.386664 | 3.102749 | 0.050591 | 0.965591 |
| | GRU | 2.393581 | 3.128216 | 0.050683 | 0.965024 |
| | LSTM-SA | 2.394424 | 3.095457 | 0.050952 | 0.965752 |
| | LSTM-TA | <u>2.104671</u> | <u>2.813940</u> | <u>0.044123</u> | <u>0.971698</u> |
| CEEMDAN + multi-network integration | RLG integration | 2.384882 | 3.102448 | 0.050459 | 0.965597 |
| | SA integration | 2.293810 | 2.974200 | 0.048594 | 0.968383 |
| | TA integration | **2.093080** | **2.808107** | **0.044101** | **0.971816** |

In Table 3, on NYtem, decomposition-based models perform significantly better than models without decomposition. In non-decomposition models, the two attention mechanisms increase the prediction error instead of decreasing it. After decomposition's import, LSTM-TA's prediction error is significantly lower than that of the reference models. For example, compared with LSTM, MAE is decreased by 11.8% and RMSE is decreased by 9.3%.

After selecting the model with the best prediction performance for each subsequence to ensemble, the prediction error is reduced compared with using a single model. Among them, the effect of using attention on some subsequences is better than that of all and not using attention, and the improvement of TA's prediction effect is higher than SA's. For example, compared with the single LSTM, the MAE of the SA-integrated model is reduced by 3.9% and the RMSE is reduced by 4.1%, while these of TA integration are 12.3% and 9.5%, respectively. In addition, the effects of the two attention-integrated models are also better than the RLG integration in [4].

### 5.4. Experiments on Datasets MRTS and SES
#### 5.4.1. Pre-Experiment: Global and Separate Normalization

For the multi-sequence datasets MRTS and SES, we conducted pre-experiments on two types of normalization, "global normalization" and "separate normalization", before the formal experiments. Firstly, without decomposition, the sequences after the two types of preprocessing are input to LSTM, respectively, to train and predict. The effect is shown in Table 4.

**Table 4.** Pre-experiment results of the two ways of normalization (without decomposition).

| Dataset | Way of Normalization | MAE | RMSE | MAPE | R$^2$ |
|---------|---------------------|-----|------|------|-------|
| MRTS | Global | 567.8171 | 991.9713 | 0.239498 | 0.747510 |
| | Separate | **473.0790** | **805.7689** | **0.176781** | **0.833403** |
| SES | Global | 4.599162 | 6.930198 | **0.178287** | 0.931681 |
| | Separate | **4.414155** | **6.155032** | 0.200523 | **0.946109** |

In Table 4, all the criteria for separately normalized MRTS are better than that of global normalization, while those for SES achieve the same result, except for MAPE. The primary

reason why the MAPE of separately normalized SES is higher than the globally normalized one is that the prediction error is larger for some items with smaller values.

Furthermore, for MRTS, after the sequences' two types of normalization, the partially decomposed sequences are shown in Figure 7. The specific selected subsequences are labeled on the left of the image. The comparison shows that since the lower observations in the original data account for a large part, the peaks in the obtained short and medium period subsequences are obviously biased to the higher part of the original data after its global normalization and decomposition. However, for the sequence normalized separately, the peak distribution after resolving is more even.

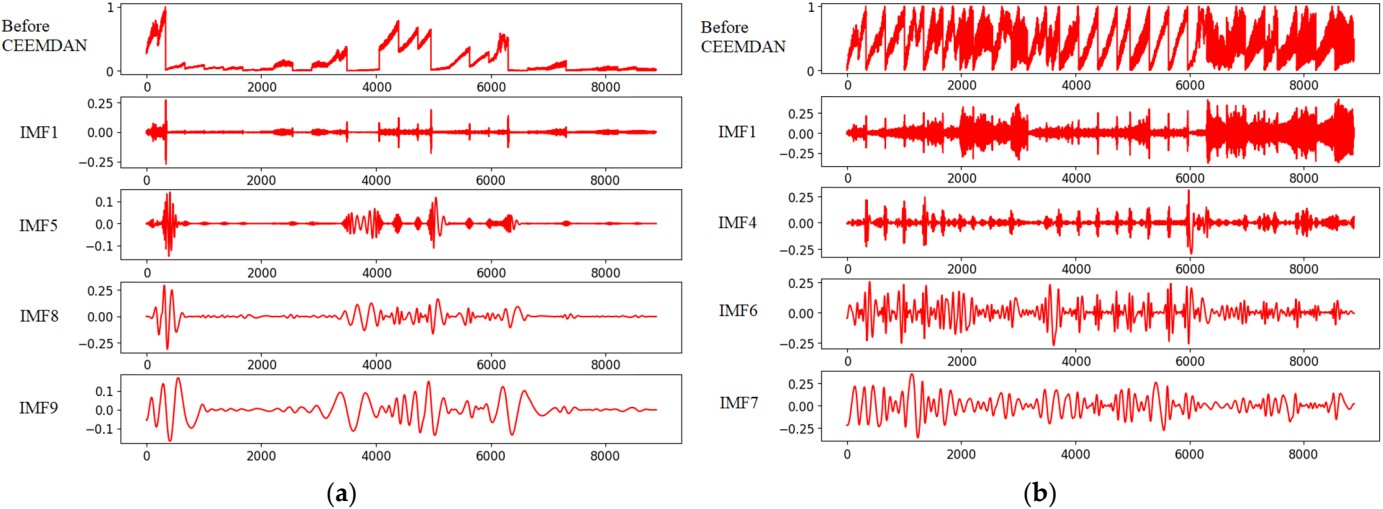

(a)        (b)

**Figure 7.** The images of the MRTS dataset and its partial subsequences after CEEMDAN. (**a**) The dataset is globally normalized; (**b**) The dataset is separately normalized.

The two sets of subsequences obtained above are input to LSTM, respectively, and the $R^2$ index comparison between them is shown in Table 5. After the decomposition of the global-normalized sequences, serious overfitting appeared on two short-term regularity sequences, $IMF_1$ and $IMF_2$, which makes the prediction of them significantly deviate from the actual results. On the contrary, the network fits much better for the sequences normalized separately.

**Table 5.** Predicting effect comparison of LSTM in MRTS between the two normalizations.

| Subsequences | $R^2$ of Global Normalization | $R^2$ of Separate Normalization |
|---|---|---|
| $IMF_1$ | $-14.070469$ | 0.467846 |
| $IMF_2$ | $-7.345910$ | 0.845353 |
| $IMF_3$ | 0.885757 | 0.981216 |
| $IMF_4$ | 0.980300 | 0.988858 |
| $IMF_5$ | 0.976094 | 0.998820 |
| $IMF_6$ | 0.965597 | 0.999776 |
| $IMF_7$ | 0.981883 | 0.999878 |
| $IMF_8$ | 0.993806 | 0.999924 |
| $IMF_9$ | 0.996590 | 0.999465 |
| $IMF_{10}$ | 0.988181 | 0.997533 |
| $IMF_{11}$ | 0.996145 | 0.999978 |
| $IMF_{12}$ | 0.962748 | N/A |
| RES | 0.999179 | 0.987735 |

According to the pre-experiment results above, both the MRTS and SES datasets will be normalized separately in the rest of the experiments. Thereafter, the experimental steps are the same as in Section 5.3, with the decomposed subsequences all normalizing secondarily as well.

### 5.4.2. Formal Experiments

After MRTS's separate normalization and CEEMDAN, 11 IMFs and a residual sequence RES are obtained. Table 6 shows the prediction results of each subsequence on LSTM and the two LSTMs with attention. As well as Table 2, SA shows superiority in short-period subsequences, while TA shows superiority in short- and medium-period ones.

**Table 6.** Prediction results of LSTM and two LSTMs with attention for each decomposed subsequence of MRTS.

| Subsequences | LSTM | | | LSTM-SA | | | LSTM-TA | | |
|---|---|---|---|---|---|---|---|---|---|
| | **MAE** | **RMSE** | **$R^2$** | **MAE** | **RMSE** | **$R^2$** | **MAE** | **RMSE** | **$R^2$** |
| $IMF_1$ | 0.068942 | 0.097606 | 0.467846 | 0.065848 | 0.092235 | 0.524803 | **0.057779** | **0.080788** | **0.635440** |
| $IMF_2$ | 0.037138 | 0.050192 | 0.845353 | 0.035867 | 0.048719 | 0.854293 | **0.026625** | **0.038382** | **0.909564** |
| $IMF_3$ | 0.005232 | **0.010196** | **0.981216** | 0.006584 | 0.010877 | 0.978625 | **0.005231** | 0.010331 | 0.980717 |
| $IMF_4$ | 0.001694 | 0.003324 | 0.988858 | 0.008016 | 0.011720 | 0.861490 | **0.000901** | **0.001767** | **0.996852** |
| $IMF_5$ | 0.000490 | 0.000722 | 0.998820 | 0.001642 | 0.002323 | 0.987764 | **0.000186** | **0.000271** | **0.999833** |
| $IMF_6$ | 0.000398 | 0.000536 | 0.999776 | 0.002089 | 0.002830 | 0.993748 | **0.000209** | **0.000286** | **0.999936** |
| $IMF_7$ | 0.000545 | 0.000696 | 0.999878 | 0.002323 | 0.003289 | 0.997277 | **0.000268** | **0.000344** | **0.999970** |
| $IMF_8$ | 0.000947 | 0.001202 | 0.999924 | 0.004963 | 0.006195 | 0.997991 | **0.000263** | **0.000341** | **0.999994** |
| $IMF_9$ | 0.000995 | 0.001297 | 0.999465 | 0.001978 | 0.002493 | 0.998023 | **0.000688** | **0.000879** | **0.999754** |
| $IMF_{10}$ | **0.000416** | **0.000526** | **0.997533** | 0.001791 | 0.002043 | 0.962785 | 0.002017 | 0.002209 | 0.956496 |
| $IMF_{11}$ | **0.000159** | **0.000224** | **0.999978** | 0.000552 | 0.000700 | 0.999784 | 0.001272 | 0.001806 | 0.998562 |
| RES | **0.000607** | **0.000729** | **0.987735** | 0.003025 | 0.003572 | 0.705902 | 0.002139 | 0.002418 | 0.865267 |

For MRTS, we select two integrated models below according to Table 6:

- SA integration: LSTM-SA is used for $IMF_1$ and $IMF_2$, while LSTM is used for the others.
- TA integration: LSTM-TA is used for $IMF_1$, $IMF_2$, and $IMF_4 \sim IMF_9$, while LSTM is used for $IMF_3$, $IMF_{10}$, $IMF_{11}$, and RES.

The results are listed in Table 7. The characteristics reflected between Tables 3 and 7 have many similarities. The main difference is that, for the undecomposed sequence, the two attention mechanisms play a certain role in improving the prediction accuracy, especially TA. After decomposition is introduced, the improvement effect of TA is significantly higher than that without decomposition, whose decline degree of MAE is increased from 2.1% to 17.3%, and that of RMSE is raised from 3.5% to 16.1%.

The experiment was carried out on SES following the same contents and steps as MRTS, and the results are shown in Table 8. The characteristics presented in the table are generally the same as the experiments on the previous two datasets, and LSTM-TA and TA integration with decomposition still have obvious advantages in prediction accuracy. However, there is almost no difference between the effect of optimal ensemble and all using LSTM-TA.

**Table 7.** Results of the experiment on the MRTS dataset.

| Type | Network Model | MAE | RMSE | MAPE | $R^2$ |
|---|---|---|---|---|---|
| Without CEEMDAN | SVR-linear | 526.0849 | 969.2392 | 0.218535 | 0.758949 |
| | SVR-RBF | 463.5917 | 817.6718 | 0.176396 | 0.828445 |
| | XGBoost | 457.5255 | 819.7784 | 0.166906 | 0.827560 |
| | LightGBM | 454.6330 | 828.9988 | 0.167352 | 0.823658 |
| | BP | 515.1523 | 962.2463 | 0.212070 | 0.762415 |
| | CNN | 516.4068 | 955.7559 | 0.213203 | 0.765609 |
| | RNN | 529.6522 | 931.3066 | 0.217999 | 0.777448 |
| | LSTM | 464.5581 | 766.3151 | 0.173935 | 0.849318 |
| | GRU | 475.4387 | 766.6409 | 0.180633 | 0.849190 |
| | LSTM-SA | 460.4295 | 823.7485 | 0.167469 | 0.825885 |
| | LSTM-TA | 454.5888 | 731.5010 | 0.170132 | 0.862698 |
| CEEMDAN + single network | RNN | 471.7071 | 717.2306 | 0.205216 | 0.868003 |
| | LSTM | 386.4517 | 574.5975 | 0.169103 | 0.915282 |
| | GRU | 360.5774 | 536.4187 | 0.158428 | 0.926167 |
| | LSTM-SA | 367.6106 | 546.2686 | 0.163798 | 0.923430 |
| | LSTM-TA | <u>319.7796</u> | <u>481.9345</u> | <u>0.136828</u> | <u>0.940403</u> |
| CEEMDAN + multi-network integration | RLG integration | 360.7650 | 536.8711 | 0.158490 | 0.926042 |
| | SA integration | 358.9588 | 534.3266 | 0.160022 | 0.926741 |
| | TA integration | **319.0218** | **480.0937** | **0.136684** | **0.940858** |

**Table 8.** Results of the experiment on SES dataset.

| Type | Network Model | MAE | RMSE | MAPE | $R^2$ |
|---|---|---|---|---|---|
| Without CEEMDAN | SVR-linear | 4.220502 | 6.381952 | 0.182274 | 0.942063 |
| | SVR-RBF | 4.109615 | 6.111284 | 0.178699 | 0.946873 |
| | XGBoost | 4.397779 | 6.321137 | 0.192321 | 0.943161 |
| | LightGBM | 4.365276 | 6.217920 | 0.189436 | 0.945003 |
| | BP | 4.414531 | 6.233717 | 0.188793 | 0.944723 |
| | CNN | 4.442804 | 6.231893 | 0.194008 | 0.944755 |
| | RNN | 4.354441 | 6.101125 | 0.191706 | 0.947049 |
| | LSTM | 4.489087 | 6.286058 | 0.210149 | 0.943791 |
| | GRU | 4.395441 | 6.142638 | 0.198855 | 0.946326 |
| | LSTM-SA | 4.421953 | 6.132696 | 0.196389 | 0.946500 |
| | LSTM-TA | 4.337513 | 6.047821 | 0.197218 | 0.947970 |
| CEEMDAN + single network | RNN | 2.687751 | 3.686859 | 0.119663 | 0.980664 |
| | LSTM | 2.677974 | 3.646671 | 0.117467 | 0.981083 |
| | GRU | 2.680054 | 3.668520 | 0.120776 | 0.980856 |
| | LSTM-SA | 2.665125 | 3.683923 | 0.116307 | 0.980695 |
| | LSTM-TA | **2.398506** | <u>3.588900</u> | **0.097303** | <u>0.981678</u> |
| CEEMDAN + multi-network integration | RLG integration | 2.671909 | 3.690628 | 0.118131 | 0.980624 |
| | SA integration | 2.601956 | 3.604627 | 0.109592 | 0.981517 |
| | TA integration | <u>2.400186</u> | **3.583131** | <u>0.097391</u> | **0.981737** |

*5.5. Analysis*

5.5.1. Analysis of the Results

According to the experiment results on the three datasets shown in Tables 2, 3 and 6–8, no matter what neural network model is selected, the prediction error of the models using EMDs is 11%~45% lower than that without decomposition. Comparing the prediction effects of each decomposed subsequence, the prediction error of SA for the short-term regularity sequences is reduced, but by less than 5%, while that of TA is 15%~28%. Meanwhile, the effect of most medium period subsequences is also improved, with mostly more than 46% of error reduced, while the highest is 72.2%. From the final effect, in the model using EMDs, the prediction accuracy after adding TA to some or all subsequences is significantly higher than that without attention or by adding SA, while MAE drops by 10%~17% com-

pared with no attention and 10%~13% compared with adding SA. The effect of the model with TA is also significantly better than that of classical neural networks such as RNN and GRU. In a word, when TA is applied to some subsequences after decomposition, the prediction effect is significantly greater than that without decomposition, without attention, and with SA. What is more, multi-sequences after separate normalization achieve forecast effects as good as single subsequences.

From the experiment results, the subsequences in which TA plays an advantage are concentrated in short and medium periods. However, it is also found in the experiments that at the junction of the short- and medium-term subsequences, a sequence with an abnormal effect appears, whose prediction effect in LSTM is better than that with TA. Since there is little diversity between the TA-integrated model and the single TA, in practical applications, we can just disable attention to long-term regularity subsequences and add TA to the others, which is known as "TA' integration". The effect comparison between TA' integration, TA integration, and all the models with TA is shown in Table 9. The table shows that there is little interval between the effect of TA- and TA'-integrated models.

**Table 9.** Comparison of training results between two TA-integrated models and LSTM-TA.

| Dataset | Model | MAE | RMSE | MAPE | R$^2$ |
|---|---|---|---|---|---|
| NYtem | LSTM-TA | 2.104671 | 2.813940 | <u>0.044123</u> | <u>0.971698</u> |
| | TA integration | <u>2.093080</u> | **2.808107** | **0.044101** | **0.971816** |
| | TA' integration | **2.092645** | <u>2.808982</u> | 0.044135 | 0.971798 |
| MRTS | LSTM-TA | 319.7796 | 481.9345 | 0.136828 | 0.940403 |
| | TA integration | **319.0218** | **480.0937** | **0.136684** | **0.940858** |
| | TA' integration | <u>319.3912</u> | <u>481.1079</u> | <u>0.136731</u> | <u>0.940608</u> |
| SES | LSTM-TA | <u>2.398506</u> | 3.588900 | <u>0.097303</u> | 0.981678 |
| | TA integration | 2.400186 | **3.583131** | 0.097391 | **0.981737** |
| | TA' integration | **2.396544** | <u>3.585854</u> | **0.097185** | <u>0.981709</u> |

### 5.5.2. Analysis of the Model Performance

For each neural network model applied in decomposed time series, we counted the total number of training parameters and the time cost to train the same epochs on the current experiment environment. Meanwhile, two types of Transformer, TransFormer encoder (TFencoder) and TransFormer encoder with Feed-Forward decoder (TF-FF), are introduced to compare.

Taking the MRTS dataset as an example, the results are shown in Table 10. According to Table 10, compared with LSTM, the prediction error of LSTM-TA is reduced by 17% at the cost of a 54% increase in time cost and a two-fold increase in space cost. In fact, this performance of LSTM-TA is still relatively better in the current environment. Under our experiment environment and the same training epochs, the time cost of TFencoder is 14.4 times that of LSTM or 8.65 times that of LSTM-TA, while that of TF-FF is 17.2 times that of LSTM or 10.3 times that of LSTM-TA. The total number of training parameters of TFencoder is 221 times that of LSTM or 76 times that of LSTM-TA, while that of TF-FF is 338 times that of LSTM or 117 times that of LSTM-TA.

**Table 10.** The training effect and performance comparison of neural network models (on MRTS dataset).

| Neural Network Model | Relative MAE with Decomposition (LSTM = 1) | Relative RMSE with Decomposition (LSTM = 1) | Total Number of Training Parameters | Training Time Cost (s/100 Epochs) |
|---|---|---|---|---|
| RNN | 1.220611 | 1.248231 | 1153 | 9 |
| LSTM | 1.000000 | 1.000000 | 4480 | 12 |
| GRU | 0.933046 | 0.933556 | 3393 | 15 |
| LSTM-SA | 0.951246 | 0.950698 | 7585 | 17 |
| LSTM-TA | **0.827476** | **0.838734** | 12993 | 20 |
| TFencoder | N/A | N/A | 989607 | 173 |
| TF-FF | N/A | N/A | 1515777 | 206 |

Moreover, experiments on an open-source code (https://github.com/BorealisAI/scaleformer, accessed on 3 November 2023) show that the number of Transformer's training parameters is 4.86 times that of TF-FF with the same properties or 567 times that of LSTM-TA (7366145). The amount of Informer's [32] and Autoformer's [33] training parameters is close to Transformer's parameters, while their training time costs are, respectively, 1.26 and 1.09 times that of Transformer's. Compared to Transformer, the space and time costs of Reformer [31] both exhibit a 60% reduction (the amount of its training parameters is 2893825); however, this improvement still leaves a considerable gap between Reformer and LSTM-TA. Consequently, we believe that under the experimental environment of this paper, the proposed model achieves relatively good prediction results and maintains good overall performance.

In addition, the training time cost we counted is only for a single decomposed sequence. Since each subsequence is relatively independent, if conditions permit, all subsequences can be trained in a parallel environment, in which case the total training time will only depend on the subsequence and model with the longest time cost. Due to the limitation of the experiment environment, only theoretical analysis is carried out here.

## 6. Conclusions and Prospect

In time-series prediction, EMD is a type of method that generates subsequences and separates short-term tendencies from long-term ones. Since the studies of EMD are limited to single sequences, our model is also used in multiple sequences with a discussion of preprocessing by "global normalization" and "separate normalization". On the other hand, the prediction effect possibly has diversity when the same model is applied to different subsequences. To obtain the regularities of subsequences more accurately with the attention mechanism, we propose an integrated time-series prediction model based on signal decomposition and two attention mechanisms, namely self-attention and temporal attention. Experiments on various datasets show that, compared to the model without attention, temporal attention increases the prediction accuracy of short- and medium-term decomposed series by, respectively, 15%~28% and 45%~72%, and the total prediction error is reduced by 10%~17%. The integrated model with temporal attention reduced the error by approximately 0.3%, compared to using attention in all subsequences. When multiple sequences are normalized separately, the prediction effect is equivalent to that of single sequences.

The datasets were selected from some common areas and have a certain representativeness. Consequently, the proposed model can be applied to most common single and multiple sequences. What is more, the performance comparison between neural network models proves that our model achieves relatively good prediction results and as maintains good overall performance in the current environment. Compared with Transformer and its variants, our model has lower space overhead and faster training speed, which is more suitable for running in parallel environments with low space requirements.

Although CEEMDAN improves the traditional EMD and can obtain subsequences with more accurate periodic features, its time cost is still much larger than EMD. In the future, we will consider more sequence decomposition strategies to optimize our model.

On the other hand, due to the constraints of time and experiment environment, only the attention mechanism and single-step prediction are considered in this research, and we have only theoretically analyzed the feasibility of parallelism. If conditions permit, we will consider the following factors:

1. Introducing more mainstream neural network models, such as Transformer and its variants, to research their application in decomposed sequences and the optimal integration problem.
2. Multi-step prediction as appropriate.
3. Parallel training of the subsequences. Moreover, emerging computing technologies, such as edge computing and cloud computing [46], can be considered to enable remote data processing and efficient learning with prediction, thereby enhancing the overall effect and performance.

**Author Contributions:** Data curation, X.W. and R.Z.; investigation, X.W.; methodology, X.W., S.D. and R.Z.; project administration, X.W.; resources, X.W.; software, S.D.; validation, S.D. and R.Z.; writing—original draft, S.D.; writing—review and editing, X.W., S.D. and R.Z. All authors have read and agreed to the published version of the manuscript.

**Funding:** This research received no external funding.

**Data Availability Statement:** The datasets used and/or analyzed during the current study are all publicly available on Kaggle: (NYtem) https://www.kaggle.com/datasets/sudalairajkumar/daily-temperature-of-major-cities (accessed on 3 April 2023). (MRTS) https://www.kaggle.com/datasets/landlord/usa-monthly-retail-trade (accessed on 24 March 2023). (SES) https://www.kaggle.com/datasets/sdorius/globses (accessed on 11 November 2022).

**Conflicts of Interest:** The authors declare no conflict of interest.

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
