# Peer review of "An Integrated Time Series Prediction Model Based on Empirical Mode Decomposition and Two Attention Mechanisms"

_information, doi:10.3390/info14110610_

Round 1

Reviewer 1 Report

Comments and Suggestions for Authors

The paper is quite interesting and yields good results.
The only comment I have is to add more models to the evaluation. It is quite common to use CNNs to explore time-series data, so the authors should also consider them for the evaluation.

Comments on the Quality of English Language

The paper requires only minor revisions.

Reviewer 2 Report

Comments and Suggestions for Authors

This paper presents an integrated model for time series prediction using Empirical Mode Decomposition (EMD) and two attention mechanisms. It combines outputs from three networks: LSTM, LSTM with self-attention, and LSTM with temporal attention, trained on EMD sub-sequences. Results show that temporal attention significantly boosts accuracy for short- and medium-term predictions, reducing total prediction error by over 10%. The integrated model with temporal attention outperforms other attention-based models, reducing error by about 0.3%.

  • A stand-alone Related Work section is needed. In fact, there are various studies on time series prediction. The authors should consider reporting at least recent works on the context. Some pointers that the authors could consider citing are [1,2], which are reported at the end of the review. I also suggest introducing a table to summarize the most recent works and to highlight the novelty of the proposed work.
  • What are the limitations of the proposed study?
  • The discussion of the results should better highlight the novelty of the proposed study and the evaluation performed.
  • Although the method is described soundly and the experimental evaluation is solid, the implications are not very clear. I suggest the authors to provide a discussion about the theoretical and practical implications of such work.
  • The quality of language could be improved, for the benefit of journal readers.
  • While you outline potential future research directions, expanding on specific research questions or challenges in these areas would provide more guidance to researchers interested in pursuing these topics.

Suggested references:

[1] Enhanced air quality prediction by edge-based spatiotemporal data preprocessing. Computers & Electrical Engineering, 96, 107572. (2021)

[2] A multisensor data fusion algorithm using the hidden correlations in Multiapplication Wireless Sensor data streams. In 2017 IEEE 14th International Conference on Networking, Sensing and Control (ICNSC) (pp. 96-102). IEEE. (2017)

Reviewer 3 Report

Comments and Suggestions for Authors

Overview

This paper considers time series prediction based on a hybrid procedure, which entails three steps including "disinteegration - prediction - integration". In particular, at the disintegration step, Empirical Mode Decomposition (EMD) is applied to original signal to extract the IMFs; subsequently, each IMF is modeled through some neural network that consists of LSTM/LSTM-SA/LSTM-TA modules. Finally, the predictions of the IMFs are combined to reach the final prediction of the original signal. 

The idea of performing time series prediction leveraging the decomposition of the original signal is not new to the community. In recent years, the idea of first decomposing the signal followed by (sub)-prediction models using neural networks have been explored many times; in particular, there are published research articles that exactly leverage EMD as the decomposition module, with the very task being performing forecasting (see, e.g., [1,2,3]). The exact neural network module in subsequent steps may not necessarily be identical to the one considered in this paper, however, at the framework level, the novelty behind this paper seems somewhat limited. 

Finally, it is also worth noting that the overall idea is closely related to the broader community that considers multi-scale problems; for example, one recent work within the time series prediction context is given in [4]; however, this large community and their related work has not been mentioned at all, despite the close connection. 

In summary, the authors seem to have ignored many of the closely-related existing literature altogether, and the literature review is very inadequate. 

Specific Comments

Please find my specific comments in the attached referee report. Please also find the above-mentioned references in the attached pdf. 

Comments on the Quality of English Language

Although the manuscript is comprehensible, many of the sentences read strange.

E.g., "Time series has various characteristics such as periodicity, stationarity and uncertainty. General time series not only is unstable in the trend but also has obvious fluctuation in the micro level, i.e., the change of data has strong uncertainty between adjacent sampling time points" -- it's not difficult to guess what message sentence is trying to convey, but the English sentence behind these are somewhat awkward (for one, uncertainty is not a CHARACTERISTIC of time series, it's a general concept that is applicable to many settings and problems). 

To this end, I suggest the authors to do a thorough language editing. 

Round 2

Reviewer 2 Report

Comments and Suggestions for Authors

The authors successfully addressed almost all my concerns. The authors deliberately ignored the suggested references, although they correctly address the considered problem and are related to the work strictly. Nonetheless, the authors did not provide any rebuttal on why the suggested references have not been considered.

In light of this consideration, I suggest discussing how the proposed approach could be applied in the work of [1] and what benefits it could provide.

[1] Enhanced air quality prediction by edge-based spatiotemporal data preprocessing. Computers & Electrical Engineering, 96, 107572. (2021)

Reviewer 3 Report

Comments and Suggestions for Authors

I would like to thank the authors for taking the time to revise the manuscript. The quality of the work has improved significantly compared with the previous version. 

The only drawback that I am seeing is that during the benchmarking analysis, the authors seem to have ignored the state-of-the-art (SOTA) methods developed by the DL-time series forecasting community over the past few years all together.

All the benchmarking methods, especially the NN ones, seem to be the very "baseline" ones as opposed to anything competitive. For example, for LSTM (with potential add-ons), given the presence of Transformer, why using LSTM-SA (which has only a single attention head), as opposed to the "real" transformer? LSTM-SA may serve as an entry for an ablation study, however, is far from enough for proper benchmarking, to see how well the decomposition + separate prediction framework can do better than some powerful forecasting tools that are on the vanilla non-decomposed time series. 
